# Characterization of Burning Behaviors and Particulate Matter Emissions of Crop Straws Based on a Cone Calorimeter

**DOI:** 10.3390/ma14123407

**Published:** 2021-06-20

**Authors:** Wei Song, Yanrong He, Yuzhang Wu, Wei Qu

**Affiliations:** Research Institute of Wood Industry, Chinese Academy of Forestry, Beijing 100091, China; sw@caf.ac.cn (W.S.); 18600991247@163.com (Y.H.); wyz@caf.ac.cn (Y.W.)

**Keywords:** crop residue, burning behavior, particulate matter, cone calorimeter, emission characteristics

## Abstract

Crop residue burning is one of the major sources of particulate matter (PM) in the air. The burning behaviors and PM emissions of the three typical crop residues (rice straw, wheat straw, corn straw) in China were characterized by a cone calorimeter (CONE) coupled with a laser dust meter. The water-soluble compounds, carbonaceous content, and morphology of PM were measured by ion chromatography, elemental analyzer, transmission electron microscopy (TEM) and energy-dispersive X-ray spectrometer (EDS). The results showed that thermal stability of corn straw was the worst among the three crop straws. The heat release rate (HRR) curves of the three crop straws were the typical curves of thermally thick charring (residue forming) samples. Wheat straw had the highest smoke yield, which was 2.9 times that of rice straw. The PM emission factor of wheat straw was 180.91 µg/g, which was about three times that of rice straw. The contents of K^+^, Na^+^, and Cl^−^ in PM were significantly higher than those of the other six water-soluble inorganic ions. The ratio of organic carbon and elemental carbon (OC/EC) ranged from 14.82 to 30.82, which was similar to the results of open burning. There were mainly three kinds of aggregates in the PM of crop straws: network, chain-like, and soot. Individual particles were mixtures of KCl and organic matters. Core-shell structures were found in PM of rice straw and corn straw. The results in this study were provided based on CONE, an ISO-standard apparatus, which could avoid data conflicts caused by the difference of combustion devices. The relationship between the burning behavior and PM emission characteristics of crop straws was established, which is helpful to understand emissions of crop straws and to find a novel way to solve the problems from the burning of crop residues.

## 1. Introduction

The open burning of crop residue occurs frequently after harvesting in China. Crop residue is also widely used for domestic cooking and heating as renewable energy. It was estimated that 600 Tg crop residue is produced every year in China, and approximately 140 Tg is burned [1]. Crop residue burning is one of the major sources of fine particles which play an important role in regional and global climate and human health [2]. Recent studies of crop residue burning have focused on the calculation of emission inventories and chemical composition of particulate matter (PM). Qiu and co-workers [3] developed a high-resolution emission inventory from open biomass burning in China. Li and co-workers [4] analyzed the morphology, composition, and mixing state of individual particles emitted from crop residue burning. However, these results are not always consistent and comparable. For example, the results of the size distributions and chemical compositions of wheat straw PM based on the online dilution system [4] differ from the results based on the residential stove [5]. Fine and co-workers [6] compared the emission characteristics of particulate matter produced by wood burning in fireplaces and wood stoves. Compared with burning in fireplaces, the yield of PM in wood stoves was smaller. The variability in particulate emissions reported in the literature depends not only on the sampling procedure but also arises from the use of different fuels, combustion technologies and operating conditions. At present, most of the devices used for simulating biomass burning are non-standard equipment and self-designed. It is necessary to find a standard combustion device to more accurately acquire the emission characteristics of particulate matter.

As an ISO-standard device, the cone calorimeter (CONE) is widely used to characterize the burning behaviors of various materials. The CONE can monitor burning behavior in real time and obtain some parameters on heat and smoke, which have a good correlation with real fire [7]. Fateh and co-workers [8] measured the combustion characteristics of two types of plywood based on CONE. Reisen and co-workers [9] characterized the emission of VOC and PM under different radiation intensities of 10 common materials for construction and furniture based on CONE. This indicates that CONE is a useful experimental device for the measurement of the gaseous and particulate emissions during combustion.

A large proportion of particles generated from biomass burning are composed primarily of PM [10]. The chemical composition of PM has an important impact on the atmospheric environment and ecosystem. In order to understand the emission characteristics of fine particles more comprehensively, various kinds of methods [11,12] (e.g., aerosol mass spectrometer (AMS), ion chromatograph (IC)) can be used to acquire the mass concentration and chemical composition of particles emitted from different sources. In addition, individual particle methods can further provide detailed information about their morphology and composition [13]. However, the relationship between biomass burning behavior and emission characteristics of PM have been not well established.

In this study, the three typical crop residues (rice straw, wheat straw, and corn straw) in China were burned by CONE. The concentration of PM was measured during combustion by a laser dust meter. The water-soluble compounds, carbonaceous content, and morphology of PM were analyzed with transmission electron microscopy (TEM) and energy-dispersive X-ray spectrometer (EDS). The main objectives of this study are to: (1) explore the burning behaviors and emission characteristics of PM from biomass combustion based on CONE, which can establish a standard method to avoid the conflicts between different studies; and (2) establish the relationship between crop residue burning behavior and PM emission characteristics, which can provide a novel way to reduce the risk of PM.

## 2. Materials and Methods

### 2.1. Materials

Three typical crop straws of rice (*Oryza sativa* L.), wheat (*Triticum aestivum* L.), and corn (*Triticum aestivum* L.) were collected from Lianyungang regions in the Jiangsu province. All samples were stored in the same room (20 °C, RH = 65) for two weeks before the tests. The results of industrial analysis and elemental analysis are shown in Table 1 and Table 2.

### 2.2. Thermal and BurningTests

#### 2.2.1. The Thermal Test

Thermogravimetric analysis (TGA) of crop straws was performed by using the SDT Q600 thermal analyzer (TA Instruments, New Castle, DE, USA). The samples with a 3 ± 1 mg weight were placed in an alumina crucible and at a heating rate of 10 °C/min under an air atmosphere from 30 to 600 °C.

#### 2.2.2. Burning Test and PM Collection

The crop straws were burned by CONE (C3, TOYO SEIKI, Tokyo, Japan) under well ventilated conditions. The heat source radiation intensity was 50 kW/m^2^ with a spark as the igniter, which is described in ISO 5660 [14]. Tests were carried out under atmospheric air and the flow rate in the CONE exhaust was equal to 24 L/s. In Scheme 1, a laser dust instrument (LD-5H, Jiuzhoupengyue Technological Co., Ltd., Beijing, China) was connected with the CONE to monitor the concentration of PM during combustion. During the experiment, the following parameters were determined: time to ignition (TTI), heat release rate (HRR), peak heat release rate (PHRR), total heat release (THR), extinction coefficient, and total smoke production (TSP).

The sampling port was located 0.8 m into the chimney flue. Straw samples (60 g per sample) were cut into 100 mm sections and placed in the 100 × 100 × 35 mm^3^ stainless steel holders, clamped with stainless steel mesh. The photographs of sample holders are shown in Figure 1b. The sampling interval and a total collection of LD-5H were 10 s and 900 s, respectively. The sampling flow rate was equal to 2 L/min. PM from the burning tests was collected on the quartz fiber filter (Ø 47 mm, Whatman, Little Chalfont, UK). In order to eliminate volatiles and impurities, the quartz filter was baked at 550 °C for 2 h before each test. In addition, before and after sampling, the filter was weighted, placed in a desiccator with a constant temperature and relative humidity (RH = 65, 20 °C) for 24 h, and stored in a −4 °C refrigerator for analysis. Three replications were performed for each crop straw.

### 2.3. Measurements of PM

#### 2.3.1. Water-Soluble Inorganic Compounds

The water-soluble inorganic compounds (WSIC) of PM were measured by using ion chromatography (LC-20ADsp, SHIMADZU, Kyoto, Japan). One quarter of quartz filter was extracted with deionized water (10 mL) for 20 min, and the extractant was used for analysis. The cation and anion columns were Ion Pac CS12A and AS14A, respectively. An eluent solution mixture, Na_2_CO_3_ (8 mol/L) and NaHCO_3_ (1 mol/L), was used for anion separation (flow rate of 1 mL/min), and methane sulfonic acid (20 mol/L) was used for cation separation (flow rate of 1 mL/min). The detected WSIC mainly includes Na^+^, NH_4_^+^, K^+^, Ca^2+^, Mg^2+^, F^−^, Cl^−^, NO^3−^, and SO_4_^2−^. The extractants were filtered using microporous membranes (0.22 mm pore size) to remove any insoluble materials [15].

#### 2.3.2. Carbonaceous Components

The carbonaceous components of PM were analyzed by using another quarter of quartz filter which was made of high-purity quartz fiber (SiO_2_). This material is suitable for analyzing the organic carbon (OC) and elemental carbon (EC) content of PM, because of high temperatures, its excellent structural stability (900–1000 °C), high purity and low background value. The mass concentrations of total carbon (TC) and OC in the sample were determined by the elemental analyzer (vario EL cube, Elementar, Heraeus, Germany). TC and OC were measured based on different temperature gradients. The quartz filter was heated to 450 °C with oxygen for 10 min to measure the OC, and heated to 1000 °C to measure TC wrapped in aluminum paper [16]. EC could be calculated as the difference TC and OC [17].

#### 2.3.3. Morphological and Elemental Analysis

The morphologies of PM samples were analyzed by a transmission electron microscope (TEM) at 300 kV (Tecnai G2-F30, FEI, Hillsboro, OR, USA) equipped with an energy-dispersive X-ray spectrometer (EDS). As for the sample preparation, the copper mesh was placed in the front of the filter during combustion. The elemental composition of PM was semi-quantitatively analyzed by EDS. Due to the interference of the copper TEM grid substrate, the Cu peak in the EDS spectrum was not analyzed.

## 3. Results and Discussion

### 3.1. Thermal Properties of Crop Straws

The TGA curves of the crop straws under the air atmosphere are shown in Figure 2a. The curves of rice straw and wheat straw have similar trends while the curve of corn straw is always below them. This indicates that the thermal stability of corn straw is the worst among the crop straws. At 500 °C, the straws are decomposed completely. The residue ratios are 18.87% of rice straw, 17.02% of wheat straw, and 9.75% of corn straw. Rice straw has the highest residue ratio, which is consistent with the results of ash content in Table 1. Figure 1b shows the differential thermogravimetric (DTG) profiles for all samples. The DTG profile shows two distinct decomposition regions which are thought to exist due to the initial volatile combustion and char burnout. The maximum rate of weight loss is found to fall between 250 and 350 °C in the first step. The thermal decomposition of cellulose is primarily thought to occur between these temperatures. The shoulder-like peaks at about 200 °C are considered to be partly related to hemicellulose within the samples. The thermal decomposition of hemicellulose took place at lower temperatures in comparison to cellulose [18]. Corn straw appears to have the largest second step at 420 °C with a maximum rate of decomposition. This reflects the lignin content of the samples. Ghetti and co-workers [19] determined a correlation between lignin content and extent of the second thermal decomposition region on the DTG profile.

### 3.2. Burning Behavior of Crop Straws

The volatile substances in the crop straws consist mainly of low-molecular hydrocarbons which are easier to ignite [20]. In Table 2, there are no significant differences between the ignition times of the three straws (about 7–10 s). The HRR curves of crop straws are shown in Figure 2a. HRR is a very important parameter for evaluating the burning behaviors of biomass materials [21]. The shapes of the HRR curves of the three crop straws are similar. They are the typical curves of thermally thick charring (residue forming) samples [22]. At the beginning, HRR increases initially until an efficient char layer forms. As the char layer thickens, HRR decreases. From Figure 2a and Table 3, the PHRR of corn straw is 167 kW/m^2^ and occurred at 13 s, which is the highest and fastest among the three crop straws. This indicates that corn straw is easier to burn and undergo pyrolysis, which is consistent with TGA. This is attributed to the higher C and H contents of corn straw. The HRR and THR of rice straw is lowest because of its highest ash content and its lowest net calorific value. The high ash content leads to aggregation and limits heat and mass transfer, resulting in incomplete combustion [23].

The curves of extinction coefficient with time (K curve) reflect the concentration of smoke during combustion, as shown in Figure 3a. At the beginning of combustion, there is a strong smoking process for the three crop straws, because of the burning of flammable volatiles. In the subsequent combustion, there are still multiple strong smoking processes for wheat straw and corn straw while there is no more smoking process for rice straw. Organic matter, carbon suspended particles and vapors are the main substances in the formation of smoke. The difference in smoke production of crop straws may be related to the difference of composition [24]. TSP of rice straw, wheat straw, and corn straw at 900 s are 0.28 m^2^, 0.81 m^2^, and 0.75 m^2^, respectively. Wheat straw has the highest smoke production which is 2.9 times that of rice straw. In addition, the later increase of smoke production is mainly due to the smoldering combustion [25].

### 3.3. PM Emission Characteristics

#### 3.3.1. PM Mass Concentration and Emission Factor

The PM mass concentrations of three crop straws were monitored in real time by the LD-5H laser dust meter during the burning process, as shown in Figure 4a. There are multiple peaks in the curves of PM mass concentrations of wheat and corn straw while there is only one main peak in the curve of rice straw. The shapes of curves of PM mass concentrations are similar with that of the extinction coefficient, indicating that PM mass concentration changes with the smoke production. After the ignition, the smoke temperature increased due to the heat release. With oxygen consumed rapidly, more PAHs, aliphatic compounds, carbonyl compounds, terpenoids, and other organic volatiles were produced. They were more prone to undergo surface reactions and to aggregate under the conditions of the saturated vapor pressure to form particulate matters [26]. There are some peaks in the curves of PM mass concentration in the smoldering stage, which is attributed to the nucleation and condensation of volatiles under incomplete combustion [27,28]. The emission factor is calculated through the PM mass and the sample mass loss. In Figure 4b, the PM emission factor of wheat straw is 180.91 µg/g, which is about three times that of rice straw. The results of PM emission factor are consistent with TSP. Thus, methods used to reduce smoke production can also reduce PM emissions.

#### 3.3.2. Water-Soluble Inorganic Compounds (WSIC) of PM

The relative contents of WSIC in PM are shown in Figure 5. The contribution of total WSIC to PM is 16.07–31.65%, which is consistent with the results of previous studies [5,10]. The total content of WSIC of rice straw is the highest (31.65%), while that of corn straw is the lowest (16.07%). This is mainly caused by the difference of composition of crop straws [29]. Between PM emitted from the burning of crop straws, the relative contents of the nine WSIC are slightly different. However, the contents of K^+^, Na^+^, and Cl^−^ in PM are significantly higher than those of the other six water-soluble inorganic ions. The sum of the three ions account for 90.02%, 83.75% and 79.46% of the total WSIC of rice straw, wheat straw, and corn straw, respectively. This may be due to the use of chemical fertilizers and herbicides at the soil sampling site. Therefore, the three ions can be used as PM source trackers for biomass combustion [30]. The order (from high to low) of relative contents of K^+^ in PM, 12.25% of wheat straw, 10.04% of rice straw, and 9.76% of corn straw, is the same with the order of the K content of the crop straws in Table 2. The same results are found on Na^+^ and Cl^−^. This indicates that the chemical composition of PM is related closely with the composition of the crop straws. Compared with coal combustion, the lower contribution of SO_4_^2−^ in PM (0.98–1.35%) may be due to the relatively low sulfur content in the crop straws [31].

#### 3.3.3. Carbonaceous Components of PM

The contents of OC, EC and TC in PM of crop straws and the OC/EC ratios are shown in Table 4. The results show that TC content in PM is 45.08–56.37%. OC in PM is derived from incomplete combustion reactions, while EC is mainly generated at higher temperature under complete combustion [32]. The difference in carbonaceous components of PM (OC and EC) is not related to only the chemical composition of the crop straws but also the condition of combustion [24]. TC contents in PM of crop straws are close to that of the residential bituminous coal combustion (ca. 45%) [33], while they are higher than that of the residential anthracite combustion (ca. 17%) [34]. OC contents in PM are higher than that of samples collected from Chinese urban atmosphere (ca. 17–20%), while the EC contents are close [35]. These results show that the OC/EC ratio in the aerosol of biomass combustion is higher than that in atmospheric aerosol. The OC contents are 36.77% of rice straw, 41.24% of wheat straw, and 44.68% of corn straw, which are less than that of the three crop straws measured by the in-house designed online dilution system [5]. This indicates that the time of flame combustion is longer and the smoldering time is shorter under CONE at 50 kW/m^2^ compared to the in-house designed system. The OC/EC ratio ranged from 14.82 to 30.82, which is close to the results of open burning [10]. The OC/EC ratio is often used for source tracking, and it is believed that the OC/EC ratio of different emission sources is different [36]. In this study, the OC/EC ratios are from 14.82 to 30.82, which could be preliminarily used as an important indicator to determine the burning of crop straws.

Moreover, total contents of WSIC and carbonaceous components in PM from the three crop straws are 76.73%, 72.47%, and 72.44%, respectively. These results are close to that of the experiment in open burning (68.9–78.9%) of Chuang and co-workers [10], but lower than the results of Chen and co-workers [5] (more than 80%). This difference may be related to the nature of the fuel and the combustion conditions. However, all the results indicate that carbonaceous components and WSIC are the main components of PM emitted from the crop straw combustion.

#### 3.3.4. Morphological and Elemental Analysis of PM

The sizes of PM emitted from the combustion of crop straws were roughly counted by TEM, as shown in Figure 6. It shows that PM is mainly ultra-fine particulate matter. The particle size distributions are consistent with previous reports on biomass burning (0–150 nm) [37]. Li and co-workers [5] found that the average geometric diameter of the particles produced by the combustion of crop straws was 75–120 nm under open burning experiments. There is 82.02% of the PM of wheat straw in the range of 1–40 nm, which is 2.38 times and 1.32 times that of rice straw and corn straw, respectively. However, most of the PM of rice straw is mainly in the range of 40–80 nm (62.78%). The proportion of rice straw PM in 80–120 nm is higher than that of wheat straw PM and corn straw PM. This indicates that the combustion of rice straw prefers to produce relatively larger particles, which might be related to the nucleation and aggregation of PM with the lower HRR and C and H contents.

The morphology and element compositions of PM from the crop straw burning were measured by TEM-EDS. As shown in Figure 7, there are mainly three kinds of aggregates in the PM of rice straw: network (Figure 7a), chain-like (Figure 7b), and soot (Figure 7c). The individual particles are spherical or irregular (Figure 7d–f). These are consistent with the results of PM from biomass combustion studied by Torvela and co-workers [38]. The individual particles aggregate and coalesce by colliding with each other, with the arrangement being more orderly [27]. The soot aggregate is composed of ultra-fine particles which are mainly organic matter, doped with KCl, according to the corresponding EDS spectra. The formation of soot particles is due to the incomplete combustion of straw at higher temperatures [4]. It can be observed that an individual particle contains one or more cores, which are formed by chemical reactions and agglomeration of PAHs produced during combustion [26]. From the EDS results of [2,3], the intensity of KCl in cores is higher than the intensity of KCl in shells. Potassium salts and organic matter are the predominant species in the PM.

As shown in Figure 8, the network and soot aggregates were found in the PM of wheat straw while there was no chain-like morphology. The soot is also composed of ultra-fine particles. However, the KCl contents of particles in soot is different. According to EDS spectra, the proportion of KCl is higher where it is darker in TEM images. The morphology of individual particles is uniform spherical, without an obvious core. This indicates that the individual particles are mixed by organic matters and KCl uniformly. However, there is the difference of surface roughness between particles in Figure 8c. This might be due to the adsorption of secondary particles on the surface of particles [39]. The intensity of Si in rough surface particles is higher than that of smooth surface particles.

The morphologies of PM from corn straw combustion are also mainly network and soot, as are those of wheat straw. However, the soot of corn straw is almost entirely composed of organic matter. This might be related to more volatiles and higher HRR because of the hemicellulose pyrolysis in the beginning of combustion. The morphology of individual particles is core-shell spherical, which is similar with that of rice straw PM. There are multi-cores in one particle in Figure 9c, indicating that the aggregation of some small particles occurred in the cooling process through the organic matter shell. There are some darker euhedral particles in Figure 9a, which might be the KCl crystals [27].

## 4. Conclusions

Crop straws are widely used for cooking and heating in rural regions of China. After harvesting, the crop residues are commonly burned in open agricultural fields. This combustion is estimated to be one of the largest sources for emissions of OC and EC in all of China. In this study, the burning behaviors of rice straw, wheat straw and corn straw were explored based on a CONE calorimeter and the chemical composition and morphology of PM emitted from the combustion of crop straws were characterized.

We summarized some rules based on our study: (1) The thermal stability of corn straw is the worst among the crop straws. The residue ratios are 18.87% of the rice straw, 17.02% of the wheat straw, and 9.75% of the corn straw at 500 °C. (2) HRR curves of the three crop straws are the typical curves of thermally thick charring (residue forming) samples. The PHRR of corn straw is 167 kW/m^2^ and occurred at 13 s, which is the highest and fastest among the three crop straws. This is attributed to the higher C and H contents of corn straw. TSP of rice, wheat, and corn straw at 900 s are 0.28 m^2^, 0.81 m^2^, and 0.75 m^2^, respectively. Wheat straw has the highest smoke yield which is 2.9 times that of rice straw. (3) The trend of PM mass concentration and emission factor are similar to the extinction coefficient and THR of crop straws, respectively. The emission factor of wheat straw is 180.91 µg/g, which is about three times that of rice straw. (4) Total contents of WSIC and carbonaceous components in PM from the three crop straws are 76.73%, 72.47%, and 72.44%, respectively. The total content of WSIC of rice straw is the highest (31.65%), while that of corn straw is the lowest (16.07%). The contents of K^+^, Na^+^, and Cl^−^ in PM are significantly higher than those of the other six water-soluble inorganic ions. The OC/EC ratio ranged from 14.82 to 30.82, which are close to the results of open burning. (5) There are mainly three kinds of aggregates in PM of crop straws: network, chain-like, and soot. The chain-like aggregates are only found in PM emitted from the combustion of corn straw. Individual particles are mixture of KCl and organic matters. Core-shell structure is found in the PM of rice straw and corn straw.

The results in this study are provided by CONE, an ISO-standard apparatus, which can avoid data conflicts caused by the difference of combustion devices. The results from different groups can be compared with each other. Meanwhile, the relationship between the burning behavior and PM emission characteristics of crop straws was established, which is helpful to understand emissions of crop straws. According to the relationship, the elemental transformation between the straws and PM could be changed, and a novel way to solve the PM issue might be invented.

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
