# Peer review of "Characterization of Burning Behaviors and Particulate Matter Emissions of Crop Straws Based on a Cone Calorimeter"

_materials, 2021, doi:10.3390/ma14123407_

Round 1

Reviewer 1 Report

The authors looked at burning residue of crops in China.  The authors did a nice job characterizing the particles matter.  The TEM work in the paper is very nice.  In the conclusions, is there alternatives that could be used than burning the straws.  Can you comments on the effects to human health?  The morphologies are nano-size, we don't know what breathing in nanoparticles does to the human lungs.  Doesn't the burning of these crop pose a huge human health risk.

  1. Can you define 600 Tg? What does Tg stand for here?
  2. Can you define the limit of detection the Cone can detect at?  Is this limit of detection within emission limits set by world environmental agency for particular matter.
  3. Sub-heading can you capitalized all the words
  4. Make sure all abbreviations are define.  Towards the results, there are so many abbreviations it's hard to keep up.  Maybe reduce the amount of words you abbreviate

Author Response

1. In the conclusions, is there alternatives that could be used than burning the straws.

Answer: Thanks a lot for the reviewer’s good question. There are some alternative methods to deal with the straws. For example, the straws could be prepared into the straw fiberboard. The straw can be a source of cellulose by enzymatic hydrolysis. The straw can also be pressed to obtain a particle fuel. However, the collection and transportation of straws are limitation of the above methods. Now, the burning of straws is still the lowest cost and the most convenient method for farmers.

2. Can you comment on the effects to human health? The morphologies are nano-size, we don't know what breathing in nanoparticles does to the human lungs. Doesn't the burning of these crop pose a huge human health risk.

Answer: Thanks a lot for the reviewer’s good questions. Many papers are published about the effects of PM of straws to human health. In No.2 reference of this paper, the health risk is reviewed. As the particles can easily penetrate deeply into lungs, they increase the frequency and severity of asthma attacks. The penetration of PM can thus aggravate bronchitis and other lung diseases while reducing the body's ability to fight infections (Trasande and Thurston, 2005). In animal experiments, the contamination status of PM-bound transition metals (e.g., chromium, cobalt, copper, manganese, nickel, titanium, vanadium, zinc, etc.) was strongly correlated with the extent of radical activation and lung injury (Kleinman et al., 2007).

3. Can you define 600 Tg? What does Tg stand for here?

Answer: Thanks a lot for the reviewer’s good questions. Tg is a unit of weight which means teragram, 1012 g.

4. Can you define the limit of detection the Cone can detect at? Is this limit of detection within emission limits set by world environmental agency for particular matter.

Answer: We greatly appreciated the reviewer’s question. In this study, the detection by CONE is measured under good ventilate, which might not simulate the real open burning perfectly. However, CONE that can adjust the oxygen concentration is already applied in other studies.

5. Sub-heading can you capitalized all the words

Answer: We greatly appreciated the reviewer’s reminder. The sub-heading has been capitalized.

6. Make sure all abbreviations are defined. Towards the results, there are so many abbreviations it's hard to keep up. Maybe reduce the amount of words you abbreviate

Answer: We greatly appreciated the reviewer’s reminder. All abbreviations have been revised.

Reviewer 2 Report

I do not have any remarks. The article is well written.

Author Response

We greatly appreciated the reviewer’s positive comments. 

Reviewer 3 Report

Dear Authors,
below, please find several remarks, about what issues in my opinion can be improved in your manuscript:
- line 35 - "regional" instead of "reginal"
- line 52 - "widely" instead of "wildly"
- line 70 and more - please, specify more clear the aim of the paper
- line 80 - when you list the tested materials, please provide a latin names of crops you tested
- line 102 and equipment parameters description - If the air/smoke flow through chimney was forced (for example by fan), please provide the flow speed
- line 171 - 174 - in my opinion this paragraph can be moved to "Methodology" chapter

Regards!

Author Response

1. line 35 - "regional" instead of "reginal"

Answer: We greatly appreciated the reviewer’s reminder. This corresponding word has been changed in the revision.

2. line 52 - "widely" instead of "wildly"

Answer: We greatly appreciated the reviewer’s reminder. This corresponding word has been changed in the revision.

3. line 70 and more - please, specify more clear the aim of the paper

Answer: We greatly appreciated the reviewer’s requirement. The aim of the paper needs more clear statement. Thus, the corresponding sentences have been changed into “The main objectives of this study are to: (1) explore the burning behaviors and emission characteristics of PM from biomass combustion based on CONE, which can establish a standard method to avoid the conflicts between different studies; (2) establish the relationship between crop residue burning behavior and PM emission characteristics, which can provide a novel way to reduce the risk of PM.”.
4. line 80 - when you list the tested materials, please provide a latin names of crops you tested.

Answer: We greatly appreciated the reviewer’s requirement. The latin names of crops have been added. Thus, the corresponding sentences have been changed into “Three typical crop straws of rice (Oryza sativa L.), wheat (Triticum aestivum L.), and corn ( Triticum aestivumL.), were collected from Lianyungang regions in the Jiangsu province.”.

5. line 102 and equipment parameters description - If the air/smoke flow through chimney was forced (for example by fan), please provide the flow speed

Answer: We greatly appreciated the reviewer’s requirement. The flow speed of CONE is 24 L/s. And the sampling flow rate was equal to 2 L/min. These have been added in the revision.

6. line 171 - 174 - in my opinion this paragraph can be moved to "Methodology" chapter

Answer: We greatly appreciated the reviewer’s recommendation. The corresponding sentences have been moved.

Reviewer 4 Report

The paper can be considered for publication after minor reviews.

  • The abstract contains some acronyms that can be explained (e.g., HRR, OC/EC).
  • The methodology and results are clear and well described.
  • You should consider adding a "Discussion" section to describe how the paper and its results respond to the first question that you set in the "Introduction" section. In fact, you should describe how the study and results contribute to the issues in the literature related to the creation of a consistent PM emission inventory. In this section, you could highlight how to standardize results with this aim.
  • Conclusions are a summary of results but they don't explain how the study and results can be used in further work.

Author Response

1. The abstract contains some acronyms that can be explained (e.g., HRR, OC/EC).

Answer: We greatly appreciated the reviewer’s reminder. These corresponding sentences have been changed into “The heat release rate (HRR) curves of the three crop straws were the typical curves of thermally thick charring (residue forming) samples.” “The ratio of organic carbon and elemental carbon (OC/EC) ranged from 14.82 to 30.82, which was similar with the results of open burning.”.

2. You should consider adding a "Discussion" section to describe how the paper and its results respond to the first question that you set in the "Introduction" section. In fact, you should describe how the study and results contribute to the issues in the literature related to the creation of a consistent PM emission inventory. In this section, you could highlight how to standardize results with this aim.

Answer: We greatly appreciated the reviewer’s reminder. As the reviewer said, the contribution of this paper to the PM issue should be discussed more deeply. Thus, these sentences have been added in Results and Discussions as follows: “The results of PM emission factor are consistent with TSP. Thus, methods used to reduce smoke production can also reduce PM emissions.”.

3. Conclusions are a summary of results but they don't explain how the study and results can be used in further work.

Answer: We greatly appreciated the reviewer’s reminder. The influence of this study to the further work is very important. Thus, the corresponding sentences in Conclusion have been changed into “The results in this study are provided by CONE, an ISO standard apparatus, which can avoid data conflicts caused by the difference of combustion devices. The results from different groups can be compared with each other. Meanwhile, the relationship between the burning behavior and PM emission characteristics of crop straws was established, which is helpful to understand how emissions of crop straws. According to the relationship, the elemental transformation between the straws and PM could be changed, a novel way to solve the PM issue might be invented.”.

Reviewer 5 Report

Comments and Suggestions for Authors

Dear Authors,

The Title:

 Characterization of burning behaviors and particulate matter emissions of crop straws based on a cone calorimeter

I have to read your manuscript with great attention and interest. The material is consistent, comprehensive and complete. The manuscript describes the behavior of crop residues - biomass during their combustion

 The submission falls within the scope of the journal and is sufficiently original, and I have a remark, so I recommended the publication after MINOR REVISIONS.

Fig. 5. is illegible, please make 3 other figures for each biomass

Fig. 6 the x-axis can be cut off to improve data visibility

Fig. 7. The test is needed, well thought out, but please arrange these graphs differently, so that it is legible and you can follow the description for the drawing with the drawing

Fig. 8. And Fig. 9- recommendations as above

Conclusion should also be prepared in points

Author Response

1. Fig. 5. is illegible, please make 3 other figures for each biomass

Answer: We greatly appreciated the reviewer’s reminder. It might be not easy to differ the results of the three crop straws in one figure. However, it is easy to compare these with each other in one sight. Apart from the three main ions, other ions are not discussed. The type size has been instead of the bigger ones to be legible.

2. Fig. 6 the x-axis can be cut off to improve data visibility

Answer: We greatly appreciated the reviewer’s recommendation. The x-axis of Fig.6 has been adjusted.

3. Fig. 7. The test is needed, well thought out, but please arrange these graphs differently, so that it is legible and you can follow the description for the drawing with the drawing. Fig. 8. And Fig. 9- recommendations as above.

Answer: We greatly appreciated the reviewer’s recommendation. These graphs and the corresponding description have been revised one by one.

Round 2

Reviewer 1 Report

Thank you for answering my questions

Author Response

Thanks a lot.